# Transcriptional Regulation of Cancer Immune Checkpoints: Emerging Strategies for Immunotherapy

**DOI:** 10.3390/vaccines8040735

**Published:** 2020-12-04

**Authors:** Simran Venkatraman, Jarek Meller, Suradej Hongeng, Rutaiwan Tohtong, Somchai Chutipongtanate

**Affiliations:** 1Graduate Program in Molecular Medicine, Faculty of Science Joint Program Faculty of Medicine Ramathibodi Hospital, Faculty of Medicine Siriraj Hospital, Faculty of Dentistry, Faculty of Tropical Medicine, Mahidol University, Bangkok 10400, Thailand; simran.ven@student.mahidol.ac.th; 2Departments of Environmental and Public Health Sciences, University of Cincinnati College of Medicine, Cincinnati, OH 45267, USA; mellerj@ucmail.uc.edu; 3Division of Biomedical Informatics, Cincinnati Children’s Hospital Medical Center, Cincinnati, OH 45267, USA; 4Division of Hematology and Oncology, Department of Pediatrics, Faculty of Medicine Ramathibodi Hospital, Mahidol University, Bangkok 10400, Thailand; suradej.hon@mahidol.ac.th; 5Department of Biochemistry, Faculty of Science, Mahidol University, Bangkok 10400, Thailand; 6Pediatric Translational Research Unit, Department of Pediatrics, Faculty of Medicine Ramathibodi Hospital, Mahidol University, Bangkok 10400, Thailand; 7Department of Clinical Epidemiology and Biostatistics, Faculty of Medicine Ramathibodi Hospital, Mahidol University, Bangkok 10400, Thailand

**Keywords:** cancer immune response, immune checkpoint inhibitor, transcription factors, tumor microenvironment

## Abstract

The study of immune evasion has gained a well-deserved eminence in cancer research by successfully developing a new class of therapeutics, immune checkpoint inhibitors, such as pembrolizumab and nivolumab, anti-PD-1 antibodies. By aiming at the immune checkpoint blockade (ICB), these new therapeutics have advanced cancer treatment with notable increases in overall survival and tumor remission. However, recent reports reveal that 40–60% of patients fail to benefit from ICB therapy due to acquired resistance or tumor relapse. This resistance may stem from increased expression of co-inhibitory immune checkpoints or alterations in the tumor microenvironment that promotes immune suppression. Because these mechanisms are poorly elucidated, the transcription factors that regulate immune checkpoints, known as “master regulators”, have garnered interest. These include AP-1, IRF-1, MYC, and STAT3, which are known to regulate PD/PD-L1 and CTLA-4. Identifying these and other potential master regulators as putative therapeutic targets or biomarkers can be facilitated by mining cancer literature, public datasets, and cancer genomics resources. In this review, we describe recent advances in master regulator identification and characterization of the mechanisms underlying immune checkpoints regulation, and discuss how these master regulators of immune checkpoint molecular expression can be targeted as a form of auxiliary therapeutic strategy to complement traditional immunotherapy.

## 1. Introduction

Cancer is a multi-faceted disease that while complex to treat possesses multiple vulnerabilities which can be targeted therapeutically. Of the several hallmarks of cancer, the study of tumor evasion has garnered interest in unraveling underlying mechanisms and identifying therapeutic targets. In recent years, there has been a rapid development of immunotherapy strategies; for example, synthetically derived chimeric antigen receptor (CAR) T cells which recognize and direct specific cytotoxicity to target cells, recombinant cancer vaccines that prompt the immune response against tumors, and immune checkpoint blockade (ICB) therapy which inhibits cancer immune evasion [1].

Immune checkpoint (IC) inhibitors have earned well-deserved appreciation for their efficacy and efficiency in combating cancer immune evasion [2]. IC inhibitors have been successfully applied across a range of cancers, consistent with the conservation of regulatory mechanisms that control immune evasion in most cancers [3,4,5,6]. The fundamental concept behind ICB is based on the reactivation of immune cells whose immune surveillance functions are curbed by expressing co-inhibitory immune checkpoint molecules [7,8]. For instance, PD-1 is a transmembrane protein expressed on T, B, and NK cells, whereas PD-L1, which belongs to the B7 family, is expressed on cancer cells [9]. The interaction between PD-1 and PD-L1 results in an interference of T-cell receptor signaling cascade, and recruits SHP-1 and SHP-2 phosphatases to tyrosine phosphorylated immunoreceptor tyrosine-based inhibitory motif (ITIM) and immunoreceptor tyrosine-based switch motif (ITSM) [10,11]. This recruitment inhibits ZAP70 and PI3K phosphorylation [12], leading to cell cycle arrest, mitigating cytokine production, and creating an immune-suppressive tumor microenvironment. When the PD-1 and PD-L1 interaction is impeded by antibody blockade, this enhances T cell functions by potentiating signal transduction from T cell receptors [9]. Conversely, an alternative strategy to modulate T cell activity in cancers is to promote the signaling of co-stimulatory molecules by using agonistic antibodies. Several co-inhibitory IC receptors have been identified for which inhibitory therapeutics have been designed, including ipilimumab for CTLA-4, Nivolumab for PD-1, Atezolizumab for PD-L1, IMP321 for LAG-3, and Epacadostat for IDO [13]. Similarly, agonists for co-stimulatory IC receptors have also been designed, e.g., Vopratelimab (JTX-2011) for the inducible T cell co-stimulator ICOS and PF-04518600 for the tumor necrosis factor superfamily receptor OX40, which are currently in phase II clinical trials [14]. However, despite the unprecedented success of ICB in multiple cancer types, less than 20–30% of patients benefit from PD-1/CTLA-4 blockade [13]. The unresponsiveness to immunotherapy due to acquired resistance represents a big and current challenge. Acquired resistance may result from genetic instability due to DNA repair alterations, higher mutational burden, neoantigen load, copy number loss of tumor suppressor genes, or upregulation of alternate co-inhibitory ICs [15]. This is reflected in a study showing a loss of PTEN promotes resistance to anti-PD-1 therapy in metastatic uterine leiomyosarcoma [16].

As the ICB resistant patient population expands, there is an increased need to overcome resistance mechanisms or circumvent it through modified forms of therapy. Much of current research is focused on either downstream effectors or upstream regulators (considered as “master regulators”) to modulate response to drug treatment as a promising avenue to overcome the resistance. This concept has been successfully applied to identify ERK and FOXM1 as master regulators of cell fate and proliferation, respectively [17,18]. As such, the master regulators of ICs, which can be defined as the transcription factors that regulate the expression of various genes involved in immune checkpoint phenomena, would hold great promise as therapeutic targets to overcome the ICB limitations. In this review, we discuss the current status and limitations of ICB therapy, summarize studies on previously identified putative master regulators of ICs, and provide an overview of emerging strategies to discover potential new therapies targeting the master regulators of ICs. Of note, this article can be considered as a narrative review and the scientific rigor of the master regulator targeting strategies should be validated in the future.

## 2. Current Issues with Immune Checkpoint Inhibitor Therapy

To shed light on ICB therapy’s limitations, we must first review briefly the interactions between tumor and immune cells. The immune response is highly regulated yet flexible and adaptive to protect the host against a myriad of pathogens. This process relies on recognizing pathogens with high sensitivity while preventing autoimmunity [7]. However, despite their often high mutational burden, cancer cells can confer resilience against immune recognition, leading to immune evasion [19]. They elicit this by limiting the expression of self-antigens, increasing co-inhibitory molecule expression [20], and down-regulating tumor antigen-presentation [21,22,23]. As stated earlier, though the ICB emergence revolutionized immunotherapy, several challenges limiting its success remain, including acquired resistance, intratumor heterogeneity, and immune-related adverse events.

### 2.1. Mechanisms of Resistance to ICB Therapy

The molecular mechanisms underlying ICB resistance have not been fully understood. However, it is generally accepted that ICB resistance is dynamic interplay between cancer cells, T cells, and tumor microenvironment (TME). One such manner in which ICB resistance is developed is by the insufficient tumor-recognizing T cells generation. This occurs by genetic or epigenetic alterations that govern the formation, presentation, and processing of tumor neoantigens, or foreign proteins unique to tumors. Changes in the signaling cascade that regulate cytotoxic T cell activity also provides an advantage to cancer cells to resist immune recognition and modulate the microenvironment to favor tumor growth and progression [15,24]. Another factor that affects ICB response is the alterations of genes encoding components of antigen processing and presentation, for instance, major histocompatibility complex—class 1 (MHC-1) and B2-microglobulin (B2M) [25]. This phenomenon generally results in acquired resistance to PD-L1 blockade, as seen in melanoma patients. In their study, the authors described that after CD8^+^ T cell infiltration during the active response, CD8^+^ T cells were abundantly present and restricted at the tumor margin, suggesting T cell-induced cytotoxicity was no longer effective [26]. This may be due to a lack of tumor antigen recognition and activation, or T cells losing their sensitivity to their effector molecules [27]. Consequently, targeting just the immune checkpoint molecules may not be sufficient to inhibit tumor immune evasions, as there are several factors, including tumor mutations, regulations of antigen presentation, modulating the TME, and even T cell response, that resulted in acquired resistance to ICB treatment [7,15].

### 2.2. Immune-Related Adverse Events of ICB Therapy

Another limitation of ICB therapy is the growing number of immune-related adverse events (irAEs). Immune checkpoint inhibitors have a different spectrum of toxicities when compared to chemotherapeutic or other biological agents, and the severity of irAEs appears to depend on immunotherapy regimens. For instance, in a phase III clinical trial that tested the safety and efficacy of atezolizumab and bevacizumab (anti-VEGF-A) plus carboplatin and paclitaxel (chemotherapeutic agents) in non-small lung cancer (NSCLC) resulted in pneumonitis [28,29,30]. This is also evidenced in a meta-analysis that compared the toxicity profile amongst different ICB drugs, including nivolumab, pembrolizumab, ipilimumab, tremelimumab, and atezolizumab. In this study, the authors revealed that even with pembrolizumab treatment, the most frequent irAE observed were pneumonitis and arthralgia [31]. This form of irAE highly corresponds to anti-PD-1 or anti-PD-L1 treatment in melanoma, NSCLC, and renal cell carcinoma [32]. On the other hand, the use of CTLA-4 inhibitors, such as ipilimumab, is highly concordant with dermatological, gastrointestinal, and renal toxicities [31]. In the treatment of melanoma, vitiligo-like depigmentation is observed. Moreover, the severe dermatological irAEs, i.e., Stevens-Johnson syndrome and toxic epidermal necrolysis, are also seen upon CTLA-4 ICB treatment [33,34]. Some of these autoimmune diseases include autoimmune pancreatitis [35], red cell aplasia [36], pancytopenia [37], and systemic vasculitis [38,39]. Of note, the irAEs that arise from anti-PD-1 are still more manageable than those from anti-CTLA-4 treatment. Further investigations should be driven towards finding ways to either mitigate the irAE with ICB therapy or devising a new therapeutic strategy that lowers the risk of developing adverse events.

## 3. Transcriptional Regulation of Cancer Immune Checkpoints

Susumu Ohno’s term, “master regulator” is defined as “a gene that occupies the very top of a regulatory hierarchy” and is not under the regulatory influence of any other gene. This term was coined to describe the regulatory mechanism of sex determination in *Drosophila melanogaster* [40]. Since its conception, the expression has evolved and found relevance in several systems, including cancer biology. Since its evolution, there have been several interpretations of the term including, “a gene that is expressed at the inception of a developmental lineage or cell type, participates in the specification of that lineage by regulating multiple downstream genes,” in terms of developmental biology [41], and “a key gene controlling cancer initiation and progression by orchestrating the associated target genes” in cancer biology. The term master regulators has also been applied to several regulatory elements, such as cis-regulatory elements, miRNAs, chromatin regulators [42], and cells as a whole [43] because these elements hold a pivotal function in gene expression and essentially dictate the outcomes of the gene regulatory network [44]. Since the ambiguity of the usage of this term may provoke confusion, this review therefore adheres to the traditional definition of the master regulators which defined as the transcription factors that regulate downstream gene expression in the pathway of interest.

### 3.1. Master Regulators in the Context of Oncoimmunology

Since master regulators hold such a pivotal role in regulating gene expression of numerous targets and thereby have implications in tumorigenesis, much effort has been focused on identifying such regulators in various cancers. In the context of ICB, this has stimulated the search for potential regulators of immune evasion mechanisms. As stated earlier, the process of immune evasion uses multiple pathways, including the modulation of TME, to generate an immunosuppressive and pro-tumoral environment. Hence, the master regulators identified for facilitating this process would have regulatory roles in multiple immune cells and may have varied functions depending on each cell type.

Consider the example of STAT3, a known master regulator of cancer immunity [45], as shown in Figure 1. STAT3 modulates the immune microenvironment by modulating the production and release of cytokines in CD8^+^ T cells and regulator T cells (Tregs), inhibiting the maturation of immature dendritic cells, and enhancing the production of ROS in Myeloid-Derived Suppressor Cells (MDSCs) [46]. STAT3 is activated by large quantities of pro-tumoral cytokines like TGF-β, Interleukin-10 (IL-10), and Interleukin-6 (IL-6). As a result, a positive feedback loop is generated in the tumoral production and secretion of these cytokines into the microenvironment. Different combinations of cytokines may behave synergistically in the activation of downstream effectors.

For instance, IL-10 and TGF-β mediate STAT3 activation and induce FOXP3 expression, which yields a positive feedback loop for the increased production of those cytokines and functions to maintain the inhibitory phenotype Tregs [47]. These cytokines also inhibit immature dendritic cell maturation, which triggers the release of IL-6 and TGF-β. [48] Together, these cytokines activate STAT3 in Th17 cells, which stimulate the production of IL-17 and MMP-2, thereby creating an immune tolerant environment for cancer cells [47,49]. On the other hand, STAT3 in CD8^+^ T cells further the immunosuppressive environment by limiting its accumulation in tumors. This is carried out by inhibiting the production or activation of CXCL10 in tumor-associated myeloid cells. In one study carried out to assess the function of STAT3 on CD8^+^ T cells, it was seen that removal of STAT3 results in an increased expression of CXCL10 receptor (CXCR3), which increases an accumulation of T-cells at the tumor site [50].

Hence, with the presence and activation of STAT3, we can infer that it holds a negative regulatory role in releasing cytokines that draw affinity towards tumor infiltration. Lastly, STAT3 is one of the main transcription factors that govern MDSC functions to promote tumor proliferation and suppress immune-mediated cytotoxic cell death of cancers [51]. MDSCs exploit the metabolic pathways via the release of ROS to limit T-cell viability and function. The release of ROS is controlled by NADPH oxidase (NOX) activity, whose expression is regulated by STAT3 [52], resulting in tumor progression through immune evasion, proliferation, angiogenesis, and metastasis [53]. Several others examples of master regulators controlling the tumor immune environment are listed in Table 1.

### 3.2. Known Putative Master Regulators of ICs

Recently, it was found that the same regulators of cell survival possess control over the expression of ICs, namely AP-1 [54], MYC [55], and ER-α [56]. This thus shows that oncogenes elicit tumorigenesis through several hallmarks of cancers as opposed to only aberrant cell proliferation and survival. AP-1, or Activator Protein 1, is a dimeric transcription factor, including members of four DNA-binding protein families (i.e., JUN, FOS, MAF, and ATF) that regulate co-inhibitory ICs molecules and is induced by the activation of co-stimulatory molecules. This is evidenced by a study conducted in mice wherein AP-1 proteins controlled the expression of IL-6 in mice after CD40 engagement [57,58,59]. Furthermore, in an in vitro study, it was observed that CD28 co-stimulation led to the activation of JNK and, therefore, the recruitment of AP-1 proteins [60,61]. Alternatively, for the regulation of co-inhibitory ICs expression, it was observed in Hodgkin’s lymphoma that AP-1 complex, consisting of c-Jun and JunB, can bind to the enhancer regions of the PD-L1 promoter [62,63]. This evidence suggests that AP-1, as a master regulator, holds a tumor-promoting role in aiding tumor cells to evade immune recognition through the enhanced expression of PD-L1. Similarly, MYC also holds a direct regulatory role in the transcription of co-inhibitory ICs, PD-L1, and CD47 [64], by binding to the promoters. However, transcription is often only stimulated when oncogenic signaling cascades, including MAPK, JAK/STAT, Wnt, and even PI3K-Akt, are activated. Typically, this is observed in oncogene addicted tumors such as lung cancer and melanoma, in which the MYC addiction of the tumors is severe enough to be a prime target of inhibition [65].

Another form of regulation by these master regulators is to enhance anticancer immunity by suppressing the expression of co-inhibitory ICs or promoting the expression of co-stimulatory ICs. This phenomenon is reflected in the negative regulation of PD-L1 by ER-α signaling in breast cancer. ER-α regulates gene transcription through the direct binding of the receptor to regulatory regions (enhancers or silencers) of target genes [56]. The relationship between ER-α and PD-L1 has been cemented in a study that revealed estrogen deprivation or ER-α depletion induced PD-L1 expression in ER+ breast cancer in vitro and in vivo [66]. While acknowledging how the regulation of IC molecular expression occurs in a bidirectional manner, we propose the master regulators of ICs as the transcription factors located at a hub of a regulatory network or as the direct transcriptional regulator responsible for the gene expression of co-inhibitory (or co-stimulatory) ICs. In this manner, AP-1 would serve as a direct transcriptional master regulator of co-inhibitory IC molecules, whereas, since there is no confirmation of a direct relationship between ER-α and PD-L1, ER-α may be located at a hub of a gene regulatory network that negatively regulates co-inhibitory ICs expression.

We summarized the list of known master regulators that elicited regulatory functions on the ICs expression, together with their broader roles in the context of oncoimmunology, in Table 1. The master regulators, as listed in Table 1, also draw attention to the knowledge gaps in thoroughly understanding the mechanism behind tumor immune microenvironment and immune checkpoint regulation, which opens an opportunity for future clinical applications. Of note, designing a therapeutic strategy to specifically target the master regulators is still challenging. Targeting these putative master regulators of ICs may result in off-target responses and increase the risk of irAEs. Furthermore, some of these regulators are seen to have contradictory roles (pro-tumoral or anti-tumoral) depending on either the combinations of certain cytokines or interactions with specific cell types. Nonetheless, the putative master regulators of ICs, as listed in Table 1, may serve as a good starting point to elucidate their potential as suitable therapeutic targets. Simultaneously, efforts should be made to identify more master regulators of ICs for fruitful results

## 4. Approaches to Identify Potential Master Regulators

To define the master regulators as the transcriptional regulators located at the hub of the gene regulatory network, one could apply a systematic identification, wherein the master regulator should be enriched for a list of differentially expressed genes (DEGs) of a particular phenotype of interest [44]. For example, Thorsson V, et al. [91] applied pan-cancer transcriptional regulatory network to describe the interactions linking genomic events to downstream target genes, immune infiltrations, and patient survival. As a result, PRDM1, SPI1, FLI1, IRF4, IRF8, STAT4, and STAT5A were identified as the key transcriptional regulators that may influence immune infiltration across cancer types [91]. While this study has given a broad perspective into key regulators of the immune response, further studies may adopt a similar approach to identify master regulators of specific immune pathways such as immune checkpoint interactions. Several methods and tools may facilitate identifying transcriptional regulators of signaling pathways and networks, such as MR4Cancer [42], iRegulon [92], and MARINa [93]. Here we listed some useful tools and their framework, which can be implemented to identify master regulators of ICs (Table 2).

## 5. Prospects on Therapeutic Targeting Master Regulators of ICs

While the discovery of master regulators does provide important insight into biological functions, to translate these findings in a clinical setting, therapeutic strategies must be devised. In the growing interest of master regulator discovery, several reports present candidate master regulators as potential therapeutic targets. This encourages further studies to investigate the therapeutic outcome of targeting these master regulators. Several strategies have been developed to harness the therapeutic potential of master regulators, including;

Modulating the expression of these master regulators to sensitize cancers to other treatmentsUsing in silico databases to find candidate small molecules to reverse signaturesDirect inhibition or stimulation of transcription factors

An example of the first strategy is seen in a study that enhances the master regulator *IKZF1*, leading to an enhanced immune infiltrate recruitment and tumor sensitivity to ICB therapy. Chen JC, et al., identified the master regulator IKZF1 that was sufficient to induce the recruitment of immune infiltrates and immune-mediated cytotoxicity in the autoimmune disease [99], which they believed representing the opposite immune signature to that of cancer immune evasion. As expected, the master regulator IKZF1 overexpression led to enhanced immune infiltrate recruitment and tumor sensitivity to PD-1 and CTLA-4 inhibitors in cancers that typically lack IKZF1 expression [100]. This study highlights the importance of modulating the expression of a master regulator to potentiate cancer immunotherapy.

As for the second strategy, a number of methods and resources have been designed to assist the identification of master regulators for a given phenotype and the enrichment of small molecules that reverse the gene expression signature. This is illustrated in the integrated transcriptomics study done in lung adenocarcinoma to identify master regulators using the ARACNe algorithm and reposition drug candidates to revert the pathological gene expression profile using the MRCMap pipeline [101]. De Bastiani MA, et al. found that the enriched drugs sheds light onto the main pathways governed by the identified master regulators and indicates the lead compound that can be repurposed to expedite drug development for lung adenocarcinoma [101]. Hence, it can be inferred that reversing the expression of the targets of the identified master regulator also returns significant results in cancer treatment.

The third strategy, wherein the master regulators are targeted directly, has reached the early phase clinical trial (NCT02716012). MTL-CEBPA is the first-in-class small activating RNA oligonucleotide drug that targets the master regulator CCAAT/enhancer-binding protein alpha (C/EBP-α) for hepatic and myeloid functions and multiple oncogenic processes [102,103,104,105]. This phase I, open-label, dose-escalation trial of MTL-CEBPA was conducted in 38 patients with advanced hepatocellular carcinoma to evaluate the safety endpoint at 28 days. The trial concluded that the use of the master regulator inhibitor MTL-CEBPA demonstrated a good safety profile, induced changes in gene expressions, as well as elicited antitumor activity [102]. Significantly, this trial adds virtue to a proposition that master regulators may be targeted as a form of therapy. By applying this concept towards identifying the master regulator of ICs and identifying small molecules to inhibit these putative regulators, we prospect that therapeutic targeting the master regulator of ICs could be a new frontier in precision immunotherapy [106].

We anticipate that implementing strategies as aforementioned to harness the therapeutic potential of master regulators may draw out the unforeseen limitations. Considering that this paradigm would be gaining traction in oncoimmunology research, there is a lack of prior studies regarding master regulators of ICs to reference and provide a foundation for new investigations. However, this knowledge gap also presents an opportunity for investigators to venture into and bridge the gap. Another drawback of adopting this new concept is the lack of available therapeutics that directly inhibit master regulators of ICs. This limitation may impede investigations from holistically perceiving the full effect master regulators have in cancer progression and encumbers translating their findings in a clinical setting. However, mechanisms can still be tested using gene manipulation techniques such as CRISPR, non-coding RNAs, and/or gene overexpression to prove the concepts, encouraging researchers and pharmaceutical companies to design therapeutics for these candidate targets. Rapid growing in the field of computational drug discovery, artificial intelligence, and big pharmacogenomic/proteomic databases and tools [107,108,109,110,111,112,113,114,115] to predict molecular targets, mechanisms of action, drug responses and adverse effects will eventually benefit the pipeline of the master regulator-targeted immunotherapeutic strategies, even though rounds of extensive benchmarking and testing in vitro and in vivo are needed before full potentials of in silico approaches can be unleashed in clinical settings. Implementation of the user-friendly, web-based programs brings a huge opportunity to scientists and clinicians knowledgeable in biology and disease-specific contexts but have less-to-no coding skills. A good example is the iLINCS (http://ilincs.org), an integrative web-based platform for the analysis of omics data and signatures of cellular perturbations stored in the Library of Integrated Network-based Cellular Signatures (LINCS) [111,112,113]. One can connect the transcriptomic (or proteomics) signatures generated from cancer cell lines with the master regulator gene knockdown (or overexpression) to those with FDA-approved (or investigational) drug treatments, aiming to sort for a list of compounds that directly act on the putative master regulators, or their major downstream effector molecules, which drive perturbagen-treated cells toward genetic manipulation-similar cellular states. With the positive results from the subsequent screening of IC molecular expression, potential therapeutic agents targeting the master regulator of ICs will be ready for further validations in preclinical and clinical phases. One might hope that further advancements in research on ICB will enable assessing the effect on individual’s transcriptome and proteome after targeting the master regulators of ICs, to recognize any off-target effects of the master regulator-targeted immunotherapy. Moreover, by aggregating these data, a pattern could be detected to predict adverse patient outcomes. Implementing such a data-centric framework requires skilled bioinformaticians to identify master regulators for patients and predict off-target effects in real-time. Lastly, this framework will require much time, effort, and capital investment to set up and run routinely in clinical practice. Nevertheless, this concept holds huge promise for personalized precision interventions in cancer treatment, and efforts must be taken to translate this paradigm into practice.

## 6. Conclusions

This review summarized the current limitations of ICB therapy and how it can be potentially circumvented by identifying and altering master regulators. The oncoimmunological functions of known master regulators are reviewed with emphasis on how these factors may regulate immune checkpoint molecule expression. While still a niche subject in oncoimmunology research, targeting the master regulators of ICs holds promise in becoming the new arm of precision immunotherapy. We further discuss possible approaches for developing therapeutic strategies around the master regulators.

## Figures and Tables

**Figure 1 vaccines-08-00735-f001:**
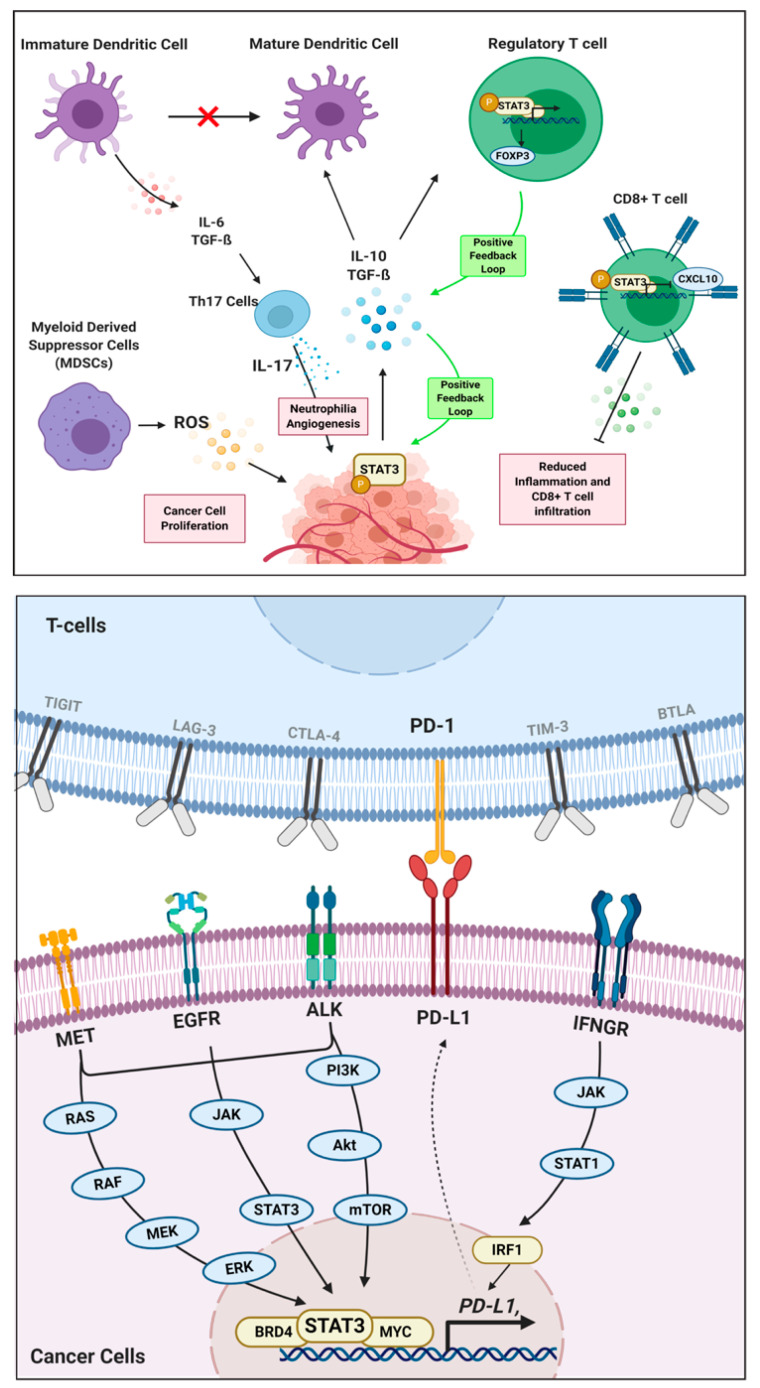
Roles of STAT3 master regulators in cancer immunity. In the context of oncoimmunology (top panel), STAT3 phosphorylation in cancer cells leads to multiple interactions between immune cells in the tumor microenvironment to promote tumor growth, angiogenesis, and metastasis. In the context of immune checkpoint regulation (lower panel), STAT3 transcription factor binds to the promoter of the co-inhibitory immune checkpoint molecule PD-L1, thereby regulates its expression and downstream effects.

**Table 1 vaccines-08-00735-t001:** List of known putative master regulators of immune checkpoint molecules.

Known Master Regulators	Roles in Immune Checkpoint Regulation	Other Roles in Oncoimmunology
AP-1	AP-1 holds a tumor-promoting role by increasing the expression of co-inhibitory ICs. In Hodgkin’s lymphoma, AP-1 response elements were identified, and cJun and JunB bind to the enhancer regions of the PD-L1 promoter. Even for co-stimulatory ICs, AP-1 binding to the promoter is required for the transcription of CD40L.	NR †
IRF-1	IRF-1 has been shown to regulate PD-L1 in various cancers, including hepatocellular carcinoma [67], pancreatic cancer [68], and melanoma [69]. This transcription factor functions by binding to the flanking region of the PD-L1 promoter. Conversely, IRF2 downregulates this transcriptional activation by binding to the response elements [67]. IRF1 has been shown to communicate with other transcription factors, including STAT1 and BRD4 [68], which affects the regulation of PD-L1. These interactions have been exploited to sensitize cancer cells to ICB therapy [70]. However, not much is known of how IRF-1 regulates other immune checkpoint molecules.	IRF-1 has been shown to have tumor-suppressive roles by activating the transcription of target genes involved in apoptosis [71]. Recent reports reveal that a subset of IRF-1 target genes, namely, LMP-2, TAP-1, MHC-1, iNOS, and IL-15, are involved in the stimulation of Immune response. This occurs through the development, expansion, and infiltration of CD8+ T and NK cells [72]s in tumors [73]. However, these roles are quite contradictory to the role IRF-1 has in regulating immune checkpoint molecules. Hence, further studies have to implore how IRF-1 elicits an evasive immune function in tumors, despite its contradictory function in stimulating cancers’ immune surveillance.
MYC	Oncogenes have been shown to regulate immune response through the modulation of PD-L1 and CD47 expression, mediated by MYC transcription factors [64]. This occurs through several growth factor receptors (EGFR and MET) [74], followed by their subsequent signaling pathways such as Beta-catenin [75], PI3K-Akt [76], and MAPK [76] signaling cascades. These signaling cascades are evidenced in melanoma and lung cancer [55].	Primarily, MYC elicits its role in regulating co-inhibitory ICs to facilitate immune evasion in cancer. However, MYC is also known to modulate the microenvironment through secreted cytokines, including thrombospondin-1 and type-1 interferon. MYC regulation of thrombospondin-1 regulates angiogenesis [77] and cellular senescence, whereas type-1 interferon influences innate and adaptive immunity [78]. Moreover, MYC also turns on immune surveillance of lymphoid malignancies via natural killer cells [65,79].
NANOG	NR †	Studies indicated that NANOG has roles in various cancers by maintaining cancer stemness, multi-modal resistance and promote metastasis and aberrant metabolism [80]. NANOG promotes an immune resistant phenotype by transcriptionally activating the Akt signaling pathway in multiple types of cancer cells [81]. NANOG also helps pancreatic cancer cells escape natural killer cell-mediated attacks by transcriptional suppression of ICAM1 [82].
STAT3	STAT3 binds to the promoter of the co-inhibitory IC antigen, PDCD1, in T-cells. Moreover, FGFR2 and EGFR expression are correlated with PD-L1 expression as FGFR2/EGFR activation stimulates STAT3 activity to transcribe PD-L1 gene [83]. This signaling cascade is observed in NSCLC [84]. Currently, the role of STAT3 in regulating other immune checkpoint molecules is not known.	STAT3 modulates the immune microenvironment by regulating the expression of cytokines. Typically for CD4^+^ T-cells, STAT3 functions in promoting proliferation and differentiation. STAT3 regulates IL-27 expression in Th1 cells, thereby increasing cell proliferation [85]. For immune tolerant T-regulatory cells, STAT3 facilitates the expression of FOXP3, which functions to maintain the inhibitory functions of regulatory T-cells [47]. Moreover, STAT3 promotes an evasive immune phenotype for cancers by inhibiting T cell expansion and cytolytic activity in hepatocellular carcinoma [86]. Lastly, STAT3 is one of the leading transcription factors that govern MDSC functions to promote tumor proliferation and suppress immune-mediated cytotoxic cell death of cancers [51].
STING	NR †	STING has antitumor roles in modulating the cytokines in the tumor immune microenvironment. It does so by facilitating the release of cancer antigens by directly triggering cell death. Additionally, activation of STING is necessary for cancer antigen presentation [87]. The activation of STING signaling in DCs results in additional protein presentation to promote T-cell activation [88]. STING also induces type-1 interferon production, which activates innate immune response against tumors [88,89]. Recently, the role of STING pathway was observed to promote immunological cell death and TME remodeling in neuroblastoma animal models [90].

† NR—Not Reported to our knowledge.

**Table 2 vaccines-08-00735-t002:** Tools for master regulator identification.

Tools	Workflow	Input	Platform	References
DIANA-miRExTra 2.0	Finds miRNA and transcription factors with crucial roles in modulating gene expression in a given gene expression data. The tool uses differential gene expression analysis and central microRNA discovery modules to predict interactions based on previously validated interactions from DIANA-TarBase.	Gene expression data	Web tool	[94] http://carolina.imis.athena-innovation.gr/mirextra/
iRegulon	Implementation of a genome-wide ranking-and-recovery approach to detect enriched transcription factor motifs and cis-regulatory elements and their optimal sets of direct targets.	A set of co-expressed genes	Cytoscape Plug-in	[92] http://iregulon.aertslab.org/
MAGIA2	An integrated analysis that uses gene expression data for reconstructing post-transcriptional gene regulatory networks. From these networks, miRNAs are identified that regulate both a transcription factor and its targets. It also identifies transcription factors that regulate miRNA and its targets. The tool integrates several miRNA databases and conducts a functional enrichment based on user-provided gene expression data	Gene expression data	Web tool	[95] http://gencomp.bio.unipd.it/magia2
MARINa	Uses gene set enrichment analysis to calculate if a gene regulatory network of a transcription factor is enriched for the DEG list provided by the user.	A set of DEGs or molecular signature and a null model	MATLAB interface	[96] http://califano.c2b2.columbia.edu/marina
MR4Cancer	A user-provided DEG list labeled with upregulation or downregulation is subjected to over-representation analysis (ORA). ORA analysis is used to assess the statistical significance of commonality between the input gene list and predetermined regulons. Increased significance indicates the likelihood of the identified MR to orchestrate the expression patterns of the input gene list.	A set of DEGs	Web Tool	[42] http://cis.hku.hk/MR4Cancer
Master Regulator Connectivity Map (MRCmap)	A transcription network inference is first drawn using gene expression data with the Bioconductor package ‘RTN’ (regulatory transcription network). This network is coupled with the master regulator analysis conducted using a two-tailed gene set enrichment analysis. This assesses the direction of inferred connection between the given master regulator and DEGs.	Phenotype contrast expression data and tissue-specific putative master regulators	R packages: RTN, Limma, PharmacoGx,	[97]
TETRAMER	Creates a gene regulatory network that includes temporal development of global transcription by integrating gene regulatory networks constructed from several transcriptomes, genome-wide mapping of promoters and enhancers in multiple cell lineages, and systemic analysis of ChIP-seq information in the NGS-QC database.	Temporal transcriptome data of two phenotypes in comparison	Cytoscape Plug-in	[98]

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
