# Peer review of "Transcriptional Regulation of Cancer Immune Checkpoints: Emerging Strategies for Immunotherapy"

_vaccines, 2020, doi:10.3390/vaccines8040735_

Round 1

Reviewer 1 Report

Dear Authors, the manuscript submitted treat a very interesting topic. However I suggest to underline that the article is a narrative review and that this could limit the scientific soundness. 

Author Response

Reviewer#1

Comments and Suggestions for Authors

Dear Authors, the manuscript submitted treat a very interesting topic. However I suggest to underline that the article is a narrative review and that this could limit the scientific soundness. 

RESPONSE: We appreciate that the Reviewer found this manuscript covers an interesting topic. We agree the Reviewer’s comment and add a statement to the main text as following;

Introduction;

Page 2, line 86-88;

“In this review, we discuss the current status and limitations of ICB therapy, summarize studies on previously identified putative master regulators of ICs, and provide a forward-looking overview of strategies to discover potential new therapies targeting the master regulators of ICs. Of note, this article can be considered as a narrative review and the scientific rigor of the master regulator targeting strategies should be validated in the future.”

Reviewer 2 Report

Reviewer’s comments to “Master regulators: Emerging concepts in cancer immune checkpoints and perspectives on new strategies of immunotherapy” by Venkatraman et al. (vaccines-1001099)

Anticancer immunotherapy has recently achieved great success in the treatment of some types of malignancies. This has become possible by pharmacologically counteracting the effect of immune-inhibitory signals emanating from tumor cells. It has become clear that naturally arising immune responses destined to destroy tumors can be specifically inhibited by various molecules produced by tumor cells, that, by binding to specific receptors on the surface of immune cells will lead to the inhibition of the activation of the immune cell and the consequent dampening of its tumor killing capacity.

The normal, physiological role of these receptor-ligand interactions is to furnish negative feedback mechanisms for lymphocyte activation in order to avoid unrestrained immune activation that would lead, for example, to autoimmune manifestations. These control mechanisms are collectively called immune checkpoints. When tumor cells acquire the capacity to express various ligands normally involved in immune checkpoint signaling, they will be protected against immune cell-induced killing. This constitutes an efficient mechanism for tumors to evade various antitumor immune responses. Inhibition of the action of tumor-derived immune checkpoint signals constitutes a promising antitumor therapeutical modality. This inhibition is currently accomplished in the clinic for example by monoclonal antibodies that bind and neutralize PD-L1, an immune checkpoint ligand, or by antibodies that bind to, and inhibit PD-1, its receptor on immune cells.  This leads to enhanced antitumor immune responses and, in a minority of cases, to curative antitumor effects, for example, is some types of malignant melanomas, which, when inoperable or disseminated, were previously incurable. In addition, and as mentioned also by the Authors, chimeric antigen receptor T cell-based therapies could also benefit from checkpoint inhibition.

Why immune checkpoint inhibition as an anticancer treatment is not universally efficient is currently unknown, and it is reasonable to hypothesize that the efficacy of this type of treatments could be enhanced if the underlying molecular mechanisms were known in greater detail and would be amenable to pharmacological modulation.

In this Manuscript Authors summarize the concept of checkpoint inhibition, discuss its limitations, and give a short overview of the current state of our knowledge regarding mechanisms that can lead to tumor resistance or escape to checkpoint inhibition therapy; adverse effects of checkpoint inhibition are discussed as well.

Authors then propose the idea of master regulators, entities that are particularly involved in the regulation of the expression of key components of immune checkpoint mechanisms. As example, STAT3 is given, and AP1, MYC and ER-a are discussed in this context. Data on the molecular mechanisms of these and other putative regulators are then discussed in an extensive Table (Table 1). This is followed by a discussion, from the standpoint of modern pharmaceutical research, on the usefulness of various available databases, and a short description of their use is described in Table 2. The usefulness of these databases is then illustrated by the identification of IKZF1 and other intracellular factors involved in intracellular signaling and transcriptional mechanisms related to immune checkpoint and its inhibition, and it is concluded that the identification of master regulators of immune checkpoints will lead to the emergence of a new arm of immune-pharmacology and therapy.

Improvement of antitumor immunotherapy based on immune checkpoint inhibition is an important goal, and an overview of the corresponding literature and of currently unsolved issues that hinder checkpoint inhibition therapy may be useful. Therefore this manuscript is timely and may, potentially interest a wide range of readers.

Comments:

1.

In the opinion of the Reviewer, the main problem with this Manuscript is that the notion of a master regulator is not defined with sufficient precision. Because the whole work is constructed upon this concept, this leads to some lack of clarity. The notion of a master regulator gene is relatively clearly defined in the context of Drosophila sex determination (as mentioned in lanes 127-129). However, the Reviewer thinks that regarding the regulation of the expression and activity of the various elements of immune checkpoint regulation, talking about such master regulators is not well funded conceptually in this Manuscript. Of course, and as stated in lanes 212-214, some specific transcription factors may indeed play important roles in the regulation of the expression of, for example PD-1 or PD-L1, and various non-genomic, epigenetic or transcriptional, or protein stability mechanisms may also be particularly important for the regulation of the mechanisms of various immune checkpoint phenomena. Authors state also that several key oncogenic mechanisms such as the activation of AP-1 (which in itself implicitly involves the entire MAP-kinase signal transduction system) and MYC (a factor that displays rather pleiotropic effects on transcription) may be master regulators, as well as ER-alpha (paragraph 3.2). This multitude of master regulators creates some confusion regarding their specificity, purpose and context-dependence. In addition it is stated (lanes 220-221) that “TIGIT… and PD-L1… have also been deemed master regulators of cancer immunity”. Would these molecules be therefore their own master regulators? Moreover, it is also mentioned that a whole cell can also satisfy the definition of a master regulator (lane 138). Given the multitude of regulatory mechanisms and biological contexts mentioned, one has the impression that the notion of a master regulator is not defined with sufficient precision. The notion of master regulators in general, and in the specific context of immune checkpoint biology need to be addressed more precisely throughout the text. The Reviewer understands that basically, Authors are interested in the identification of transcription factors that regulate the expression of various genes involved in immune checkpoint phenomena (lanes 212-214). However, this point should be made more clear earlier in the Manuscript.

Moreover, the Reviewer thinks that it is not substantiated in the Manuscript that immune checkpoint inhibition is a phenomenon that is controlled exclusively by self-autonomous mechanisms in a way similar to that defined in the context of Drosophila sex determination and outlined in lanes 127-129 (“…not under the regulatory influence of any other gene”). Thus, in the opinion of the Reviewer, reworking the entire concept of master regulators in the context of immune checkpoint regulation would add a considerable amount of necessary clarity to this Manuscript, without compromising its pertinence and information content.

2.

Lanes 21-23: “The study of immune evasion has gained a well-deserved eminence in cancer research by successfully developing a new class of therapeutics, immune checkpoint inhibitors, such as pembrolizumab and nivolumab, anti-PD-L1 antibodies.” This phrase should be reformulated, because nivolumab and pembrolizumab are not anti-PD-L1 antibodies. These antibodies are directed against PD-1, so in an immunological sense these are anti-PD-1 antibodies. Of course, by binding to PD-1, they exert an “anti-PD-L1” pharmacological effect by hindering PD-L1 binding to its receptor (i.e.: PD-1). The specificity of nivolumab is correctly stated in lane 56.

3.

In the opinion of the Reviewer, in a paper like the Manuscript submitted by the Authors, it would be appropriate to discuss the main intracellular signaling mechanisms involved in the inhibitory action of, for example PD-1, such as ITIM-dependent tyrosine phosphatase activation and its downstream consequences.

4.

Lane 299: It is not clear to the Reviewer what constitutes, in a kuhnian sense, the paradigm shift mentioned here. In general, the Reviewer thinks that the Paragraph corresponding to lanes 294-318 is too general, and conveys little useful information in its present form.

5.

As an additional note: the Reviewer believes that it would be useful for clarity to clearly and consistently distinguish in the text immune checkpoint molecules from immune checkpoint mechanisms. For example, the Reviewer believes that PD-1 is not an immune checkpoint, but an immune checkpoint molecule, an immune checkpoint being the potential or the action of PD-1 being activated by (for example) PD-L1, leading to the generation of inhibitory downstream signals in a lymphocyte. A distinction between actors and action needs to be made semantically.

Author Response

Reviewer#2

Comments and Suggestions for Authors

Reviewer’s comments to “Master regulators: Emerging concepts in cancer immune checkpoints and perspectives on new strategies of immunotherapy” by Venkatraman et al. (vaccines-1001099)

Anticancer immunotherapy has recently achieved great success in the treatment of some types of malignancies. This has become possible by pharmacologically counteracting the effect of immune-inhibitory signals emanating from tumor cells. It has become clear that naturally arising immune responses destined to destroy tumors can be specifically inhibited by various molecules produced by tumor cells, that, by binding to specific receptors on the surface of immune cells will lead to the inhibition of the activation of the immune cell and the consequent dampening of its tumor killing capacity. The normal, physiological role of these receptor-ligand interactions is to furnish negative feedback mechanisms for lymphocyte activation in order to avoid unrestrained immune activation that would lead, for example, to autoimmune manifestations. These control mechanisms are collectively called immune checkpoints. When tumor cells acquire the capacity to express various ligands normally involved in immune checkpoint signaling, they will be protected against immune cell-induced killing. This constitutes an efficient mechanism for tumors to evade various antitumor immune responses. Inhibition of the action of tumor-derived immune checkpoint signals constitutes a promising antitumor therapeutical modality. This inhibition is currently accomplished in the clinic for example by monoclonal antibodies that bind and neutralize PD-L1, an immune checkpoint ligand, or by antibodies that bind to, and inhibit PD-1, its receptor on immune cells.  This leads to enhanced antitumor immune responses and, in a minority of cases, to curative antitumor effects, for example, is some types of malignant melanomas, which, when inoperable or disseminated, were previously incurable. In addition, and as mentioned also by the Authors, chimeric antigen receptor T cell-based therapies could also benefit from checkpoint inhibition. Why immune checkpoint inhibition as an anticancer treatment is not universally efficient is currently unknown, and it is reasonable to hypothesize that the efficacy of this type of treatments could be enhanced if the underlying molecular mechanisms were known in greater detail and would be amenable to pharmacological modulation. In this Manuscript Authors summarize the concept of checkpoint inhibition, discuss its limitations, and give a short overview of the current state of our knowledge regarding mechanisms that can lead to tumor resistance or escape to checkpoint inhibition therapy; adverse effects of checkpoint inhibition are discussed as well. Authors then propose the idea of master regulators, entities that are particularly involved in the regulation of the expression of key components of immune checkpoint mechanisms. As example, STAT3 is given, and AP1, MYC and ER-a are discussed in this context. Data on the molecular mechanisms of these and other putative regulators are then discussed in an extensive Table (Table 1). This is followed by a discussion, from the standpoint of modern pharmaceutical research, on the usefulness of various available databases, and a short description of their use is described in Table 2. The usefulness of these databases is then illustrated by the identification of IKZF1 and other intracellular factors involved in intracellular signaling and transcriptional mechanisms related to immune checkpoint and its inhibition, and it is concluded that the identification of master regulators of immune checkpoints will lead to the emergence of a new arm of immune-pharmacology and therapy.

Improvement of antitumor immunotherapy based on immune checkpoint inhibition is an important goal, and an overview of the corresponding literature and of currently unsolved issues that hinder checkpoint inhibition therapy may be useful. Therefore this manuscript is timely and may, potentially interest a wide range of readers.

RESPONSE: We appreciate the Reviewer for all constructive comments and thoughtful suggestions. We are pleased that the Reviewer agree this manuscript is timely and may potentially interest a wide range of readers also. Please find below how all comments are fully addressed in a point-by-point manner.

Comments:

1. In the opinion of the Reviewer, the main problem with this Manuscript is that the notion of a master regulator is not defined with sufficient precision. Because the whole work is constructed upon this concept, this leads to some lack of clarity. The notion of a master regulator gene is relatively clearly defined in the context of Drosophila sex determination (as mentioned in lanes 127-129). However, the Reviewer thinks that regarding the regulation of the expression and activity of the various elements of immune checkpoint regulation, talking about such master regulators is not well funded conceptually in this Manuscript. Of course, and as stated in lanes 212-214, some specific transcription factors may indeed play important roles in the regulation of the expression of, for example PD-1 or PD-L1, and various non-genomic, epigenetic or transcriptional, or protein stability mechanisms may also be particularly important for the regulation of the mechanisms of various immune checkpoint phenomena. Authors state also that several key oncogenic mechanisms such as the activation of AP-1 (which in itself implicitly involves the entire MAP-kinase signal transduction system) and MYC (a factor that displays rather pleiotropic effects on transcription) may be master regulators, as well as ER-alpha (paragraph 3.2). This multitude of master regulators creates some confusion regarding their specificity, purpose and context-dependence. In addition it is stated (lanes 220-221) that “TIGIT… and PD-L1… have also been deemed master regulators of cancer immunity”. Would these molecules be therefore their own master regulators? Moreover, it is also mentioned that a whole cell can also satisfy the definition of a master regulator (lane 138). Given the multitude of regulatory mechanisms and biological contexts mentioned, one has the impression that the notion of a master regulator is not defined with sufficient precision. The notion of master regulators in general, and in the specific context of immune checkpoint biology need to be addressed more precisely throughout the text. The Reviewer understands that basically, Authors are interested in the identification of transcription factors that regulate the expression of various genes involved in immune checkpoint phenomena (lanes 212-214). However, this point should be made more clear earlier in the Manuscript. Moreover, the Reviewer thinks that it is not substantiated in the Manuscript that immune checkpoint inhibition is a phenomenon that is controlled exclusively by self-autonomous mechanisms in a way similar to that defined in the context of Drosophila sex determination and outlined in lanes 127-129 (“…not under the regulatory influence of any other gene”). Thus, in the opinion of the Reviewer, reworking the entire concept of master regulators in the context of immune checkpoint regulation would add a considerable amount of necessary clarity to this Manuscript, without compromising its pertinence and information content.

RESPONSE: We appreciate the reviewer’s comment in fully elucidating their concern about the clarity of the definition of master regulators for ICs. We acknowledge this comment by describing our definition of master regulators for ICs in lanes 231-237 and lanes 257-258. We also allude to how the examples provided (AP-1 and ER-alpha) satisfy our description. We understand how the various examples, each with their own definition of ‘master regulators’ may be confusing, however, as the topic is relatively new, we are not at liberty to dictate one definition just yet. With regards to TIGIT and PD-L1 identified as master regulators, we agree with the reviewer on how it is unclear as to what they control. In this manner, we moved that paragraph towards section 3.1 - Master regulators in the context of oncoimmunology, and further described how TIGIT and PD-L1 respectively regulate cancer immunity as following;

Main text;

Page 4, line 191 – page 5, line 199:

“Hence, with the presence and activation of STAT3, we can infer that it holds a negative regulatory role in releasing cytokines that draw affinity towards tumor infiltration. Lastly, STAT3 is one of the main transcription factors that govern MDSC functions to promote tumor proliferation and suppress immune-mediated cytotoxic cell death of cancers [52]. MDSCs exploit the metabolic pathways via the release of ROS to limit T-cell viability and function. The release of ROS is controlled by NADPH oxidase (NOX) activity, whose expression is regulated by STAT3 [53], resulting in tumor progression through immune evasion, proliferation, angiogenesis, and metastasis [54]. Several others examples of master regulators controlling the tumor immune environment are in Table 1. Of note, recent reports revealed that the immune checkpoint molecules, namely TIGIT [55] and PD-L1 [56], have also been deemed master regulators of cancer immunity.  TIGIT elicited this effect by suppressing anti-tumor immunity by mediating regulatory T-cells. Moreover, PD-L1 was shown to be upregulated in numerous immune cells, from which it either inhibits immune cell proliferation or promotes a tolerogenic tumor immune microenvironment.”

Additionally, we reworked the entire concept of master regulators in the context of immune checkpoint regulation as the Reviewer’s suggestion and stated in the main text earlier as following;

Main text;

Page 2, line 81-84:

“… regulators of cell fate and proliferation, respectively [17,18]. As such, therapeutic targeting master regulators of ICs, which defined as the transcription factors that regulate the expression of various genes involved in immune checkpoint phenomena, would hold great promise to overcome the ICB limitations. In this review, we discuss the current status and limitations of ICB therapy, summarize studies on previously identified putative master regulators of ICs, and provide an overview of emerging strategies to discover potential new therapies targeting the master regulators of ICs.”

Also, the title is changed to enhance the clarity of the manuscript as following;

Title;

Page 1, line 2-3:

Transcriptional regulation of cancer immune checkpoints: Emerging strategies for immunotherapy”

2. Lanes 21-23: “The study of immune evasion has gained a well-deserved eminence in cancer research by successfully developing a new class of therapeutics, immune checkpoint inhibitors, such as pembrolizumab and nivolumab, anti-PD-L1 antibodies.” This phrase should be reformulated, because nivolumab and pembrolizumab are not anti-PD-L1 antibodies. These antibodies are directed against PD-1, so in animmunologicalsense these are anti-PD-1 antibodies. Of course, by binding to PD-1, they exert an “anti-PD-L1” pharmacological effect by hindering PD-L1 binding to its receptor (i.e.: PD-1). The specificity of nivolumab is correctly stated in lane 56.

RESPONSE: We agree with the reviewers and apologize for our negligence. We have changed lane 24, to reflect “anti-PD-1” instead of “anti-PD-L1”.

3. In the opinion of the Reviewer, in a paper like the Manuscript submitted by the Authors, it would be appropriate to discuss the main intracellular signaling mechanisms involved in the inhibitory action of, for example PD-1, such as ITIM-dependent tyrosine phosphatase activation and its downstream consequences.

RESPONSE:  We appreciate the reviewer’s guidance on increasing the readability and understanding of this manuscript. Therefore, new statements have been added into the introduction section as following;

Introduction;

Page 2, line 53-61:

 “For instance, PD-1 is a transmembrane protein expressed on T, B, and NK cells and PD-L1 belongs to a B7 family of co-inhibitory and co-stimulatory molecules expressed on cancer cells [9]. The interaction between PD-1 and PD-L1 results in an interference of T-cell receptor signaling cascade, and recruits SHP-1 and SHP-2 phosphatases to tyrosine phosphorylated immunoreceptor tyrosine-based inhibitory motif (ITIM) and immunoreceptor tyrosine-based switch motif (ITSM) [10, 11]. This recruitment inhibits ZAP70 and PI3K phosphorylation [12], leading to cell cycle arrest, mitigating cytokine production, and creating an immune-suppressive tumor microenvironment. When the PD-1 and PD-L1 interaction is impeded by antibody blockade, this enhances T cell functions by potentiating signal transduction from T cell receptors [9].”

References;

Page 17, line 392-403:

9. Arasanz, H.; Gato-Canas, M.; Zuazo, M.; Ibanez-Vea, M.; Breckpot, K.; Kochan, G.; Escors, D. PD1 signal transduction pathways in T cells. Oncotarget 2017, 8, 51936-51945, doi:10.18632/oncotarget.17232.

10. Sharpe, A.H.; Wherry, E.J.; Ahmed, R.; Freeman, G.J. The function of programmed cell death 1 and its ligands in regulating autoimmunity and infection. Nat Immunol 2007, 8, 239-245, doi:10.1038/ni1443.

11. Chemnitz, J.M.; Parry, R.V.; Nichols, K.E.; June, C.H.; Riley, J.L. SHP-1 and SHP-2 associate with immunoreceptor tyrosine-based switch motif of programmed death 1 upon primary human T cell stimulation, but only receptor ligation prevents T cell activation. J Immunol 2004, 173, 945-954, doi:10.4049/jimmunol.173.2.945.

12. Sheppard, K.A.; Fitz, L.J.; Lee, J.M.; Benander, C.; George, J.A.; Wooters, J.; Qiu, Y.; Jussif, J.M.; Carter, L.L.; Wood, C.R., et al. PD-1 inhibits T-cell receptor induced phosphorylation of the ZAP70/CD3zeta signalosome and downstream signaling to PKCtheta. FEBS Lett 2004, 574, 37-41, doi:10.1016/j.febslet.2004.07.083.

4. Lane 299: It is not clear to the Reviewer what constitutes, in a kuhnian sense, the paradigm shift mentioned here. In general, the Reviewer thinks that the Paragraph corresponding to lanes 294-318 is too general, and conveys little useful information in its present form.

RESPONSE: We recognize that what we are proposing does not fully adhere to Thomas Kuhn's definition of paradigm shift, wherein we dispel one belief for a newer one. Rather, we applied the definition of introducing a new concept in adjacent to pre-conceived notions. Accordingly, we have changed the term "paradigm shift" in lane 319 to “new concept” as following;

Main text;

Page 14, line 319:

“Another drawback of adopting this new concept is the lack of available therapeutics …”

In addition, new statements have been added after the paragraph corresponding to lanes 294-318 as mentioned by the Reviewer to provide more useful information for the readers as following;

Main text;

Page 15, line 327-338:

“… can be unleashed in clinical settings. Implementation of the user-friendly, web-based programs brings a huge opportunity to scientists and clinicians knowledgeable in biology and disease-specific contexts but have less-to-no coding skills. A good example is the iLINCS (http://ilincs.org), an integrative web-based platform for the analysis of omics data and signatures of cellular perturbations stored in the Library of Integrated Network-based Cellular Signatures (LINCS) [121-123]. One can connect the transcriptomic (or proteomics) signatures generated from cancer cell lines with the master regulator gene knockdown (or overexpression) to those with FDA-approved (or investigational) drug treatments, aiming to sort for a list of compounds that directly act on the putative master regulators, or their major downstream effector molecules, which drive perturbagen-treated cells toward genetic manipulation-similar cellular states. With the positive results from the subsequent screening of IC molecular expression, potential therapeutic agents targeting the master regulator of ICs will be ready for further validations in preclinical and clinical phases. One might hope that …”

5. As an additional note: the Reviewer believes that it would be useful for clarity to clearly and consistently distinguish in the text immune checkpoint moleculesfrom immune checkpoint mechanisms. For example, the Reviewer believes that PD-1 is not an immune checkpoint, but an immune checkpoint molecule, an immune checkpoint being the potential or the action of PD-1 being activated by (for example) PD-L1, leading to the generation of inhibitory downstream signals in a lymphocyte. A distinction between actors and action needs to be made semantically.

RESPONSE: We thank the reviewer for this comment, and believe this will provide a much-needed clarity in immune checkpoint interactions. Therefore, we’ve specified ‘immune checkpoint molecules’ or ‘co-inhibitory molecules’, throughout the manuscript,  wherever we refer to the ‘actors’, and have left the terms immune checkpoints, whenever we referred to the ‘action’.

Reviewer 3 Report

Summary:

The manuscript entitled “Master regulators: Emerging concepts in cancer immune checkpoints and perspectives on new strategies of immunotherapy” by Venkatraman et al., is a review article submitted for publication on the special issue "Cancer Immunotherapy: Advances and Future Prospects".

The topic of the review is very interesting, timely and informative. It is well written in terms of clarity and use of English language.

Authors provides also new insights to an existing body of knowledge.

Minor issues:

We suggest  minor modifications to address the following issues to potentially improve the impact of the manuscript.

1. Recently, a putative role of the STING pathway in promoting immunogenic cell death and TME remodeling has been described in neuroblastoma patients suggesting a link between STING activation and T cell response (http://dx.doi.org/10.1136/jitc-2019-000282). Authors should consider to include this recent piece of information.

2. An emerging body of evidence suggests a key role for the MAPK-activated protein kinase 2 as master regulator of immune response in the TME.

Author Response

Reviewer#3

Summary:

The manuscript entitled “Master regulators: Emerging concepts in cancer immune checkpoints and perspectives on new strategies of immunotherapy” by Venkatraman et al., is a review article submitted for publication on the special issue "Cancer Immunotherapy: Advances and Future Prospects". The topic of the review is very interesting, timely and informative. It is well written in terms of clarity and use of English language. Authors provides also new insights to an existing body of knowledge.

RESPONSE: The authors appreciate that the Reviewer found this review is very interesting, timely and informative, and provides some useful insights to the current body of knowledge. We would like to thank for the Reviewer’s suggestions which we feel help improving the impact of the manuscript.

Minor issues:

1. Recently, a putative role of the STING pathway in promoting immunogenic cell death and TME remodeling has been described in neuroblastoma patients suggesting a link between STING activation and T cell response (http://dx.doi.org/10.1136/jitc-2019-000282). Authors should consider to include this recent piece of information.

RESPONSE: This is an important comment. As a result, this information was added into Table 1 as following;

Table 1;

Page 9; Row label – STING; Column label - Other roles in oncoimmunology:

“STING has antitumor roles in modulating the cytokines in the tumor immune microenvironment. It does so by facilitating the release of cancer antigens by directly triggering cell death. Additionally, activation of STING is necessary for cancer antigen presentation [92]. The activation of STING signaling in DCs results in additional protein presentation to promote T-cell activation [93]. STING also induces type-1 interferon production, which activates innate immune response against tumors [93,94]. Recently, the role of STING pathway was observed to promote immunological cell death and TME remodeling in neuroblastoma animal models [95].

References;

Page 23, line 637-639:

95. Wang-Bishop, L.; Wehbe, M.; Shae, D.; James, J.; Hacker, B.C.; Garland, K.; Chistov, P.P.; Rafat, M.; Balko, J.M.; Wilson, J.T. Potent STING activation stimulates immunogenic cell death to enhance antitumor immunity in neuroblastoma. J Immunother Cancer 2020, 8, e000282, doi:10.1136/jitc-2019-000282.”

2. An emerging body of evidence suggests a key role for the MAPK-activated protein kinase 2 as master regulator of immune response in the TME.

RESPONSE: We thank the Reviewer for this caption. Accordingly, information of MAPK-activated protein kinase 2 as the key regulator of immune response in the TME was added to Table 1as following;

Table 1;

Page 8; Row label – MK2; Column label - Other roles in oncoimmunology:

An emerging body of evidence suggests a key role for the MAPK-activated protein kinase 2 (MAPKAPK2 or MK2) in tumorigenesis and immune responses in the colon cancer TME [77,78]. The activation of MAPKAPK2 and its downstream mediator HSP27 in intestinal mesenchymal cells led to production of cytokines, chemokines, matrix metalloproteinases (MMPs), resulting in modulation of tumor microenvironment and promoting intestinal tumorigenesis [78].”

References

Page 22; lanes: 582-586:

77. Soni, S.; Anand, P.; Padwad, Y.S. MAPKAPK2: the master regulator of RNA-binding proteins modulates transcript stability and tumor progression. J Exp Clin Cancer Res 2019, 38, 121, doi:10.1186/s13046-019-1115-1.

78. Henriques, A.; Koliaraki, V.; Kollias, G. Mesenchymal MAPKAPK2/HSP27 drives intestinal carcinogenesis. Proc Natl Acad Sci U S A 2018,115, E5546-E5555, doi:10.1073/pnas.1805683115.

Round 2

Reviewer 2 Report

Authors addressed several issues raised by the Reviewer; a paragraph on intracellular checkpoint-related tyrosine phosphatase activation has been included, factual errors and several minor issues regarding clarity have been corrected. New text on bioinformatics is also of merit. However, a few issues still remain:

Lanes 76-77: “Much of current researcher…” : “…current research…”

Lanes 54-55: “while PD-L1 which belongs to the B7 family expressed on cancer cells…”

PD-L1 can be expressed on tumor cells, but B7 family members are normally expressed on immune cells: “…PD-L1 which belongs to the B7 family, can be expressed on cancer cells…”  ? In general, the paper needs editing for English grammar and syntax.

Lanes  81-83:”As such, therapeutic targeting master regulators of ICs, which defined as the transcription factors that regulate the expression of various genes involved in immune checkpoint phenomena, would hold great promise to overcome the ICB limitations.” : …targeting of master regulators…”,  “…which, when defined…” or “…which can be defined as…” ? Please reformulate the phrase, as it is not easily comprehensible in its present form.

Lanes 86-87: “Of note, this article can be considered as a narrative review and the scientific rigor of the master regulator targeting strategies should be validated in the future.” It is not clear what Authors intend to say in this phrase. Please note that the Review requests precisions regarding the precise definition of IC master regulators, and not the rigorous discussions of future targeting strategies.

Again, whereas in lanes 81-83 Authors define IC master genes as transcription factors : “As such, therapeutic targeting master regulators of ICs, which defined as the transcription factors that regulate the expression of various genes involved in immune checkpoint phenomena, would hold great promise to overcome the ICB limitations” However, in lanes 191-192 authors state that: “Of note, recent reports revealed that the immune checkpoint molecules, namely TIGIT [55] and PD-L1 192 [56], have also been deemed master regulators of cancer immunity”

How can these phrases be reconciled, considering that TIGIT and PD-L1 are not transcription factors ?

“Master regulators” has been deleted from the title. However, the notion of an IC master regulator remains a central notion in the entire text, while still not being defined with sufficient clarity. Is a master regulator 1) specifically a transcription factor involved in the regulation of the transcription of immune checkpoint receptors or ligands as stated in lanes 81-84, or may it be 2) such a transcription factor, as well as any other cellular molecule or phenomenon involved in the function of immune checkpoint mechanisms (for example lanes 191-199) ?

It is very disturbing for the reader when a central notion in a paper is used with changing and ambiguous definitions. The Authors have the liberty to define, what they consider an IC master regulator. However, this definition needs to be clear, and thereafter needs to be adhered to consistently, throughout the text.

As pointed out earlier, immune checkpoint molecules and immune checkpoint mechanisms needed to be distinguished in the text, and this has been done by the Authors as suggested by the Reviewer. Similarly, IC regulatory molecules (those key transcription factors discussed in the text) and regulatory mechanisms (such as signal transduction pathways or various properties of the tumor microenvironment) should be distinguished similarly. Internal contradictions and ambiguity regarding the definition of a key notion in a manuscript (master regulator) need to be eliminated.

Author Response

Reviewer#2_R2: Minor revision

Comments and Suggestions for Authors

Authors addressed several issues raised by the Reviewer; a paragraph on intracellular checkpoint-related tyrosine phosphatase activation has been included, factual errors and several minor issues regarding clarity have been corrected. New text on bioinformatics is also of merit. However, a few issues still remain:

  1. Lanes 76-77: “Much of current researcher…” : “…current research…”

RESPONSE: We apologize for this typo. This is corrected as the Reviewer’s suggestion.

Introduction;

Page 2, line 77:

“… forms of therapy. Much of current research is focused on …”

  1. Lanes 54-55: “while PD-L1 which belongs to the B7 family expressed on cancer cells…” PD-L1 can be expressed on tumor cells, but B7 family members are normally expressed on immune cells: “…PD-L1 which belongs to the B7 family, can be expressed on cancer cells…”  ? In general, the paper needs editing for English grammar and syntax.

RESPONSE: We thank the Reviewer to point this out. This sentence was corrected as following;

Introduction;

Page 2, line 54:

“…and NK cells, whereas PD-L1, which belongs to the B7 family, is expressed on cancer cells.”

  1. Lanes  81-83:”As such, therapeutic targeting master regulators of ICs, which defined as the transcription factors that regulate the expression of various genes involved in immune checkpoint phenomena, would hold great promise to overcome the ICB limitations.” : …targeting of master regulators…”,  “…which, when defined…” or “…which can be defined as…” ? Please reformulate the phrase, as it is not easily comprehensible in its present form.

RESPONSE: We agree that the phases could be improved, to which we corrected it as following:

Introduction;

Page 2, line 81-83:

“As such, the master regulators of ICs, which can be defined as the transcription factors that regulate the expression of various genes involved in immune checkpoint phenomena, would hold great promise as therapeutic targets to overcome the ICB limitations.”

  1. Lanes 86-87: “Of note, this article can be considered as a narrative review and the scientific rigor of the master regulator targeting strategies should be validated in the future.” It is not clear what Authors intend to say in this phrase. Please note that the Review requests precisions regarding the precise definition of IC master regulators, and not the rigorous discussions of future targeting strategies.

RESPONSE: This sentence was suggested by the Reviewer#1 in which we agreed to add into the previous version of the revised manuscript. The statements made here are a suggestion that the perspectives on therapeutic targeting master regulators of ICs have to be validated in the future.

  1. Again, whereas in lanes 81-83 Authors define IC master genes as transcription factors : “As such, therapeutic targeting master regulators of ICs, which defined as the transcription factors that regulate the expression of various genes involved in immune checkpoint phenomena, would hold great promise to overcome the ICB limitations” However, in lanes 191-192 authors state that: “Of note, recent reports revealed that the immune checkpoint molecules, namely TIGIT [55] and PD-L1 192 [56], have also been deemed master regulators of cancer immunity”. How can these phrases be reconciled, considering that TIGIT and PD-L1 are not transcription factors ? “Master regulators” has been deleted from the title. However, the notion of an IC master regulator remains a central notion in the entire text, while still not being defined with sufficient clarity. Is a master regulator 1) specifically a transcription factor involved in the regulation of the transcription of immune checkpoint receptors or ligands as stated in lanes 81-84, or may it be 2) such a transcription factor, as well asany other cellular molecule or phenomenon involved in the function of immune checkpoint mechanisms (for example lanes 191-199) ? It is very disturbing for the reader when a central notion in a paper is used with changing and ambiguous definitions. The Authors have the liberty to define, what they consider an IC master regulator. However, this definition needs to be clear, and thereafter needs to be adhered to consistently, throughout the text.

RESPONSE: We appreciate the Reviewer for his/her thoughtful comments and efforts towards improving our manuscript. We recognize and agree that this example in lanes 191-199 may stir confusion and contradicts with the definition in lane 81-83, and that has a negative impact to the whole manuscript.

Taken the Reviewer’s comment#6 into account, we therefore decide to redact the statements related to TIGIT and PD-L1 in lanes 191-199, and also remove TGF-b(a cytokine), mTOR and MK2 (kinases) from the main text and/or Table 1, in order to adhere to the definition in lanes 81-83. We also revise associated texts and add a new statement to emphasize the clarity and consistency of the usage of the term master regulators as the transcription factors.

Main text;

Page 4, line 147-153:

“… in cancer biology. The term master regulators has also been applied to several regulatory elements, such as cis-regulatory elements, miRNAs, chromatin regulators [42], and cells as a whole [43] because these elements hold a pivotal function in gene expression and essentially dictate the outcomes of the gene regulatory network [44]. Since the ambiguity of the usage of this term may provoke confusion, this review therefore adheres to the traditional definition of the master regulators which defined as the transcription factors that regulate downstream gene expression in the pathway of interest.”

  1. As pointed out earlier, immune checkpoint molecules and immune checkpoint mechanisms needed to be distinguished in the text, and this has been done by the Authors as suggested by the Reviewer. Similarly, IC regulatory molecules(those key transcription factors discussed in the text) and regulatory mechanisms (such as signal transduction pathways or various properties of the tumor microenvironment) should be distinguished similarly. Internal contradictions and ambiguity regarding the definition of a key notion in a manuscript (master regulator) need to be eliminated.

RESPONSE: This isa great comment. As a result, the IC regulatory molecules and regulatory mechanisms have been distinguished throughout the manuscript as the Reviewer’s guidance, while the ambiguity regarding the definition of the master regulators has been addressed together with the response to the Reviewer’s comment#5.